# Use of Drugs and Dietary Supplements in University Students of Sports Science: Results of a Survey-Based Cross-Sectional Study

**DOI:** 10.3390/nu14204267

**Published:** 2022-10-13

**Authors:** Giovanni Ficarra, Michelangelo Rottura, Pierangela Irrera, Alessandra Bitto, Fabio Trimarchi, Debora Di Mauro

**Affiliations:** 1Department of Biomedical Sciences, Dental Sciences and Morpho-Functional Imaging, University Hospital, Via C Valeria, 98125 Messina, Italy; 2Department of Clinical and Experimental Medicine, University Hospital, 98125 Messina, Italy

**Keywords:** university students, natural supplements, physical activity, survey

## Abstract

Dietary supplements are used to implement and balance common dietary habits. The general belief is that natural substances reduce the risk of chronic diseases and amplify sports performance with no harmful side effects. Since sports science students will become professionals of sport activities and may also have a role in suggesting the use of dietary supplements to athletes, the aim of this study was to evaluate if physical activity influences the use of drugs and dietary supplements. A modified version of the International Physical Activity Questionnaire—Short Form (IPAQSF) was administered to perform these evaluations. A total of 1452 students from the University of Messina, Italy, enrolled in sports science courses completed the survey; of these, 1075 (704 male and 371 female students) were physically active in moderate- or high-intensity training. Of those physically active students, 709 (440 male and 269 female) were competitive athletes, identified on the basis of their answer to a specific question also indicating the type of sport they practice. The results suggest that 5.6% of all respondents were regular consumers of pharmaceutical products, compared to just 1.0% of the cohort of competing athletes. In contrast, the consumption of natural supplements was similar (14% vs. 15%) between groups. The most frequently used supplements were vitamins, including vitamin C, vitamin B complex, and multivitamin complex, followed by minerals and amino acids or protein complex. The probability of using dietary supplements was mostly related to the male gender (OR 1.64; 95% CI: 1.17–2.30), having a job (OR 1.45; 95% CI: 1.07–1.96), and, most of all, performing physical activity (OR 3.53; 95% CI: 2.18–5.71). The only factor related to a higher use of drugs was female gender (OR 2.40; 95% CI: 1.52–3.79), and the most used class was antihistaminic, followed by FANS. These results suggest that among the specific population of sports science students, those performing physical activity are less prone to using pharmaceutical products and have healthier habits.

## 1. Introduction

Nutritional knowledge and adequate nutrient intake are essential components for enhancing athletic performance, and in this context, the prevalence of the use of nutritional (dietary) supplements, such as proteins, amino acids, carbohydrates, carnitine, creatine, and multiminerals, has increased globally [1,2].

Balanced nutrition is not always easy to achieve, especially among young people who are used to junk food and unscheduled meals. Anecdotal evidence suggests that athletes of all ages use nutritional supplements based on information primarily gathered from the internet, media, magazines, or untrained personnel. Coaches and instructors also have a role in suggesting the use of dietary supplements, but their personal background relies, most of the time, on personal experience rather than an appropriate knowledge of the benefit/harm of these substances [3]. Specifically formulated supplements for professional or amateur athletes are commonly used to boost performance and to preserve/restore the body reserves while engaged in a high level of physical activity [4]. Considering either sport-specific formulations (such as pre-workout supplements) and common supplements (such as multivitamins), another issue of concern is that they are generally acquired from supermarkets or gyms rather than from a pharmacy, without a proper evaluation of the real needs of the person and without consulting a nutrition or health specialist. Supplement use in adolescents and young adults has been reported to be mainly related to vitamin use, especially in non-elite athletes practicing individual sports at the university level [5].

In Italy, nutritional supplements are commercialized in different pharmaceutical forms (from pills to gels), but none require a doctor’s prescription and can be put on the market if the basic requirements of quality and manufacturing standards are met, according to European Regulations (Directive 2002/46/EC). This means that they do not need to be dispensed by qualified professionals who would have the knowledge to advise the individuals on the benefits and possible side effects of the products purchased. For this reason, young athletes, including those studying sports science courses at the university level, may be unaware of the possible consequences of using and over-using such supplements. Unfortunately, it is not common knowledge that many dietary components, particularly if consumed at high dosages, might have pharmacological or even toxic effects, especially if taken alongside other pharmaceutical products without informing the prescribing physician. Drugs assumed for long-term therapeutic interventions, such as hormone substitutes, cardioactive drugs, and others by athletes of any age are known to interact with dietary supplements used to improve physical performance, and generally, there is little or no knowledge of these interactions [6]. Interactions, and eventually adverse effects, may also occur with over-the-counter medicinal products that are often self-prescribed, including those used for short-term therapies, such as for pain relief or other common ailments [7].

Knowledge of the benefits and possible risks associated with the use of supplements, particularly when used concurrently with medicinal products, is important for everybody, but it is essential for future coaches and sports professionals. Such individuals in Italy generally follow specific training courses which are offered by national sports federations or similar bodies, and often follow university degree courses in the field of sports science. The curriculum in such sports science courses aims to provide students with a sound understanding of human body functions, sport methodology (i.e., how to teach a sport discipline and how to identify the appropriate training for a specific competitive athlete), and how to adapt training programs to specific categories (children, elderly, disabled, and people with various disease conditions). Most sports science courses also include classes in biochemistry and pharmacology, but such classes could be of a generic nature, and not much attention is given to the nutritional aspects of training and to the harms and benefits of supplements.

For the reasons mentioned above, it is important to ensure that the next generations of coaches have an adequate education in this specific field. Unfortunately, not much data are available on the knowledge of university students attending a sports science faculty, and no previous survey, to our knowledge, interviewed students who are likely to practice a sport and eventually pursue a coaching career and likely be asked by their clients to provide suggestions for using nutritional supplements.

The aim of the present study was to investigate the possible concomitant use of supplements and medicinal/pharmaceutical products within the specific population of university students following sports science degree courses who are more exposed to use and advise on the use of food supplements, and to analyze, through a survey, if physical activity influences the use of such medicinal drugs and dietary supplements. The findings are used to elucidate if a more specific educational approach to drug–nutrient interactions should become part of the course curriculum of sports science degree programs.

## 2. Materials and Methods

### 2.1. Participants

The survey was administered to all students enrolled in the Corso di Laurea in Scienze Motorie, Sport e Salute, a three-year sports science bachelor’s degree program at the University of Messina, Italy, in the months of June and July 2021. A total of 1780 students were exposed to the questionnaire, and 1521 agreed to participate. A total of 1452 (aged 18–30 years) completed the survey in a valid manner, and of these, 1075 (704 male and 371 female students) practiced moderate- or high-intensity training. Of those performing some kind of exercise, 709 (440 male and 269 female) were practicing a sport at a competitive level. Active students were considered as those performing physical exercise but not participating in competitions, based on their answers. The questionnaire was displayed on the student login page of the university website, and after giving consent to proceed, the students were exposed to the survey. Participation was on a voluntary basis and all questionnaires were anonymous. To maintain anonymity, all personal answers (i.e., which sport do you practice?) given by less than 3 people were not considered in the analysis. Additionally, no personal information regarding health or socio-economic status was recorded, in agreement with the policies of the Ethics Board of the University of Messina.

### 2.2. Survey

The questionnaire used was previously validated [8,9] based on its content, application, structure, and presentation (IPAQ—International Physical Activity Questionnaire). The questionnaire, following approval from the Ethics Committee of the University Hospital of Messina (Prot. 152/20 approved on 9 February 2021), was embedded on the personal university web page of the students, and the survey was self-administered online by each participant.

In this cross-sectional study, participants were asked to answer multiple-choice questions and some open-ended questions. The main questions regarded (1) the number of days of high-intensity training (HIT) in the previous month and the minutes/day (MET equivalent 8–12/h); (2) the number of days of moderate-intensity training (MIT) in the previous month and the minutes/day (MET equivalent 6–8/h); (3) use of medications; (4) use of medications in the past 7 days; (5) use of supplements during physical exercise; (6) use of supplements in the past 7 days; (7) professional who advised on the use of supplements (if any); (8) age, sex, height, weight, and job (if any); (9) sport practiced (if any); and (10) which specific drug and/or supplement they used. Questions 3 to 6 were added to our modified version, and considering their mere informative nature, were accepted by the review board. Of all the questionnaires, only those with all boxes checked were considered valid and used for analysis by the IT Centre that collected the data. Note that in Italy, it is not uncommon for university students to also have a job, and hence students were specifically asked about work they performed in addition to their studies.

MET (metabolic equivalent of task) values were attributed according to specific reference values for the sports performed by those engaged in competitive activities. Each category of activity was assigned a MET score on the energy cost, and the weighted MET-minutes per week was calculated by multiplying the standard MET score, duration, and frequency per week [10].

### 2.3. Statistical Analysis

All data were collected by the IT Centre of the University of Messina, and a coded database was sent to the investigators for the analysis. Continuous variables were summarized and reported as mean and standard deviation. Descriptive statistics were reported as absolute frequency and percentage for categorical variables, and two-tailed Pearson chi-squared tests were conducted to compare the variables.

Univariate logistic regression models were performed to identify predictors of routine drug use. All variables identified as predictors were included in a stepwise multivariate logistic regression model (backward procedure, α = 5%). Univariate and multivariate logistic regressions were also performed to identify the factors associated with the use of supplements.

Odds ratios (ORs) with 95% CIs were calculated for each covariate of interest in the univariate (crude OR) and multivariate (adjusted OR) models. The goodness of fit of the regression model was assessed by the Hosmer–Lemeshow test for adequacy. Values of *p* < 0.05 were considered statistically significant. The statistical analysis was performed with SPSS version 23.0 (IBM Corp., SPSS Statistics, Armonk, NY, USA).

## 3. Results

The university students included in the study totaled 1452, of which 914 (63.0%) were male, 906 (62.4%) were aged between 18 and 23 years, and 546 (37.6%) were older than 24 years. Furthermore, 517 subjects (35.6%) were employed and 86 of them (5.9%) had a sedentary job (the definition of sedentary work was based on the personal assessment of the participant). The high proportion of working students is related to the specific objective of this degree course—in fact, most of the students work in a sports-related facility (gym, pool, etc.). In addition, 1075 (74.0%) students performed physical activities (intended as exercise such as gym training, running, or any kind of sport), and 371 (34.5%) practiced competitive sports. In the latter group, the mean weekly MET score per person was 1800 (IQR = 1260–2880), with a mean MET score of 257.14 per day, although only some of the 371 students trained every day (n = 45) and most trained 3 to 6 days/week (n = 326). Moreover, among the physically active students, 118 (11.0%) trained every day, 266 (24.7%) trained 3 to 6 days a week, 415 (38.6%) trained less than 3 days a week, and 276 (25.7%) did not provide an answer. Usual supplement users amounted to 205 (14.1%), and multivitamin supplements were the most consumed. In particular, 178 (12.2%) had taken supplements in the past seven days (Table 1). A usual supplement user was defined on the basis of the answer to the questions regarding “use of supplements”. Furthermore, 82 (5.6%) students were regular drug users, and 8.5% (7) took more than one per day. The most used drugs were antihistamines, as shown in Table 1.

The table shows the data extrapolated from the analysis of all the records collected in the study. The most used supplements were vitamins, including vitamin C, vitamin B complex, and multivitamins, followed by minerals and amino acids or protein complex. In terms of medicinal products, the term “other drugs” was used to include all drugs taken by less than 10 people. Data concerning the routine use of drugs showed that therapies usually consist of life-saving drugs (e.g., thyroid medications) or contraceptives, associated with drugs taken for short periods such as antihistamines or antipyretics, or NSAIDs.

Statistical analysis of the data suggests that the physically active respondents had a significantly reduced use of drugs compared to those with a sedentary lifestyle (4.9% vs. 7.7%; *p* = 0.046). Conversely, a significantly higher consumption of supplements was observed in subjects who performed physical activity (17.3%) compared to those who did not (5.3%; *p* < 0.001), as shown in Figure 1A,B.

No differences in the use of drugs and supplements were observed between subjects who practiced competitive versus non-competitive sports (Figure 1C,D).

The only predictor that significantly reduced the chance of taking drugs was male gender (OR, 95% CI = 0.42, 0.26–0.66; *p* < 0.001). Physical activity reduced the probability of taking drugs but did not reach statistical significance (OR, 95% CI = 0.64, 0.40–1.04; *p* = 0.070; Table 2).

Predictive factors for the use of drugs, as reported in Table 2, using as a reference the population aged 18–20 years, appear to be unrelated to age but significantly related to female gender and having a sedentary job. The univariate analysis revealed that having a sedentary job was more related to the use of drugs (OR, 95% CI = 3.019, 1.34–6.79; *p* = 0.008), but in the multivariate analysis, only gender remained associated with the probability of taking drugs in the interviewed population.

In addition, the probability of taking supplements significantly increased in male students who performed physical activity and had a job (Table 3). On the other hand, the probability of using natural supplements, as demonstrated in the univariate analysis and reported in Table 3, was mostly related to male gender, having a job, and performing physical activity. In this case, the multivariate analysis showed that supplement use appears to be up to 3.5 times more likely in those who were more physically active.

Additionally, only 15 subjects were concomitantly taking drugs and dietary supplements in our population of 1452 students, representing 1% of the subjects but limiting the possibility of further analyses.

Only 23 subjects replied to the question regarding a “professional who advised on the use of supplements”, with “trainer” being the most frequent response (60.8%, n = 14), and none responding with “physician”. Unfortunately, the low number of respondents meant that the relative data could not be used for full analysis.

## 4. Discussion

Interviewing university students of sports science, we sought to understand if an educational intervention could be of importance to explain (i) the benefits of performing physical activity to reduce the use of drugs and (ii) the possibility of specific interactions between drugs and dietary supplements, and the results of this survey demonstrate that such an intervention should be taken. This work has, for the first time, looked into the use of supplements and/or medicinal drugs by sports science students, an area which was previously not thoroughly studied. Previously reported studies on university students mainly focused on the level of knowledge regarding nutraceuticals rather than the use of these substance or the use of medicines/pharmaceutical products in physically active students [11,12,13,14].

The results obtained in our survey show that students performing physical activity had a significantly reduced use of drugs and an increased use of dietary supplements. This correlation was independent of sex and other variables, and whether one is a competitive athlete or not. Thus, it appears clear that it may be equally beneficial, from this specific public health perspective, to encourage young people to engage in physical activity to maintain a healthier lifestyle, even at a relatively young age, rather than encouraging them to become competitive athletes. As a matter of fact, a reduced use of pharmaceutical products and an increased use of dietary supplements was observed in our “active” population and it was not dependent on any other variable. These results are in close agreement with previously published data from other surveys carried out in Italy [13] and other countries [14,15,16], although the consumption of supplements was low (14.1%) compared to previously reported data investigating university students. One possible explanation is that the survey was conducted in the period of the COVID-19 pandemic—it is well known that in countries that experienced significant contagion, including Italy, indoor physical activities as well as outdoor group activities were stopped for a long time, and this must be taken into account when comparing our results to what has previously been observed. Another factor which could have an influence is the economic collapse due to COVID-19, which has forced people to reduce their expenses, especially those related to what may be considered superfluous products [17].

Another crucial finding is the prevalence of drug use in women and sedentary people who are more prone to having chronic diseases. In our population, male sex seems to influence the attitude toward sport activity and the use of dietary supplements. These results were similar to what was previously demonstrated in another cohort from the south of Italy [18].

The differences observed among male and female students regarding the engagement in physical activity, the use of supplements and drugs, and employment are of interest considering that most of the interviewed students are young and should be more active than other students since they are enrolled in a sport-oriented degree course. Other surveys involving Croatian students of biomedical courses (not sports science) demonstrated similar gender differences, but unrelated to the use of dietary supplements [19].

The limitations of our study are its cross-sectional design and the lack of questions on the knowledge about dietary supplements. In addition, considering that the survey was administered only to students of a specific degree program (namely sports science), there is a selection bias that was intended to better understand if an educational intervention should be made for this specific population of students. Many other surveys have investigated the level of knowledge of interviewed subjects regarding supplements, but we decided to focus our attention on the possible concomitant use of drugs and supplements. Another important limitation is that the data are self-reported and memory-dependent; thus, some critical information may not have been reported. At the same time, we did not have the chance to properly investigate lifestyle, underlying diseases, and socio-economic status, since many questions were considered too sensitive by our review board. Moreover, since the students accessed the survey from their personal university webpage, we could display only a limited number of questions. In addition, considering that most of the interviewed subjects did not indicate which professional advised them on the use of supplements, it can be speculated that in most cases, supplements were self-prescribed on the basis of information taken from the internet—this must be considered a serious problem, because the interviewed students represent the next generation of coaches, and it is of outmost importance that they provide reliable advice to their future athletes.

Our findings highlight one important observation: a lack of knowledge that professional advice should be sought before taking supplements, even if these are dietary supplements that can be purchased without a prescription. In fact, although our findings suggest that in this relatively young and selected population, there is an influence of being physically active on the use of either drugs or supplements, there was an overlap between drug and supplement consumption that was probably not advised by any specialist (i.e., physician or pharmacist). More specifically, 15 subjects reported using both medications and dietary supplements, and of all the interviewed students, only 23 were advised by a professional but none by a physician. This observation supports the idea that respondents were possibly unaware of the potential interactions between the drugs and the supplements. This could be of concern, as the specific population of respondents represents an important cohort of society: the individuals who are most likely to become the sport coaches of tomorrow, i.e., the professionals who, as indicated by this study, are most likely to be consulted by athletes for advice on which supplements to take. This information highlights that a specific evidence- and science-based education should be provided on the use of dietary supplements to sports science students to help them better understand the pros and cons of dietary supplementation, and, perhaps more importantly, help them realize that their physicians should be informed of any supplement use.

## 5. Conclusions

This work suggests that in this relatively young and controlled population, being physically active influences the use of either drugs or supplements. In particular, a reduced use of drugs and an increased use of natural supplements was observed in the active population, and it was not dependent on any other variable. Practicing a competitive sport was not correlated to drug or supplement use. Another crucial result is the prevalence of drug use in women and sedentary people who are more prone to having chronic diseases. In our population, male sex seems to influence the attitude toward sport activity and the use of natural supplements. In the present investigation, we sought to understand the habits of sports science students regarding the use of supplements and medicines. Considering that these students will likely continue their career in a sport-related field, our results indicate that a deeper knowledge regarding the harms and benefits of natural supplements, especially possible contra-indications in cases of underlying diseases or chronic therapies (i.e., for epilepsy, diabetes, kidney or liver problems), is needed.

## Figures and Tables

**Figure 1 nutrients-14-04267-f001:**
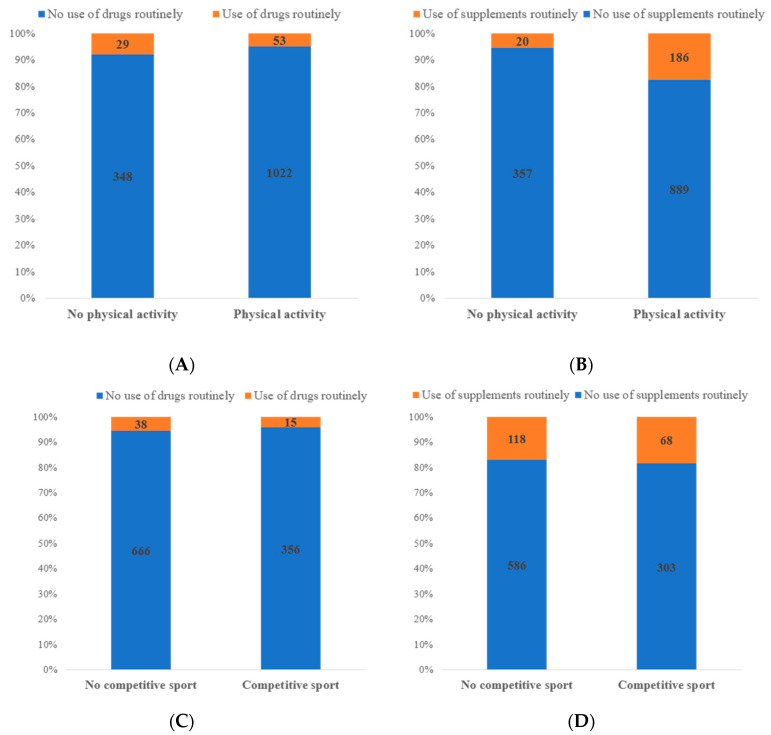
Number of subjects performing physical activity and routinely using drugs (**A**) or supplements (**B**). Number of subjects practicing competitive sports and routinely using drugs (**C**) or supplements (**D**).

**Table 1 nutrients-14-04267-t001:** Type of supplements/medicines and frequency of use in the studied population.

**Supplements**	**Frequency of Supplement Use**	
	**Past 7 Days** **178 (12.2%)**	**Usually** **205 (14.1%)**
**Vitamins**	22	11
**Multivitamins**	149	192
**Minerals/amino acids or protein complex**	5	0
**Not provided**	2	2
**Medicines**	**Frequency of Drug Use**	
	**Last 7 Days** **135 (9.3%)**	**Usually** **82 (5.6%)**
**Analgesics/Antipyretic**	18	8
**Anti-inflammatory NSAIDS**	17	8
**Antihistamines**	24	21
**Thyroid drugs**	13	13
**Other drug classes**	35	44
**Not provided**	11	6

**Table 2 nutrients-14-04267-t002:** Predictive factors of drug use according to univariate and multivariate analysis.

	Crude OR [CI 95%]	*p*-Value	Adj OR [CI 95%]	*p*-Value
**Age**				
**18–20**	----			
**21–23**	0.87 (0.38–1.98)	0.733	0.86 (0.37–2.04)	0.740
**24–26**	0.97 (0,49–1.95)	0.938	0.91 (0.44–1.87)	0.796
**27–30**	1.09 (0.52–2.29)	0.826	1.07 (0.50–2.29)	0.868
**Gender (M)**	0.42 (0.26–0.65)	<0.001	0.42 (0.26–0.66)	<0.001
**BMI**	0.96 (0.89–1.03)	0.203	0.98 (0.94–1.03)	0.458
**Physical activity**	0.62 (0.39–0.99)	0.047	0.64 (0.40–1.04)	0.070
**Competitive sport**	0.738 (0.40–1.36)	0.331		
**Sedentary job**	3.019 (1.34–6.79)	0.008		
**Other job (non-sedentary)**	0.934 (0.58–1.49)	0.776	0.98 (0.61–1.58)	0.941
**Use of supplements**	1.38 (0.77–2.47)	0.275	1.71 (0.94–3.12)	0.078

**Table 3 nutrients-14-04267-t003:** Predictive factors of supplement use according to univariate and multivariate analysis.

	Crude OR [CI 95%]	*p*-Value	Adj OR [CI 95%]	*p*-Value
**Age**				
**18–20**	----			
**21–23**	0.84 (0.50–1.42)	0.517	0.98 (0.56–1.72)	0.952
**24–26**	0.79 (0.50–1.24)	0.299	0.91 (0.57–1.46)	0.701
**27–30**	1.08 (0.67–174)	0.764	1.13 (0.69–1.86)	0.622
**Gender (M)**	1.73 (1.25–2.41)	0.001	1.64 (1.17–2.30)	0.004
**BMI**	0.99 (0.98–1.01)	0.994	0.99 (0.97–1.01)	0.393
**Physical activity**	3.735 (2.32–6.02)	<0.001	3.53 (2.18–5.71)	<0.001
**Competitive sport**	1.12 (0.80–1.55)	0.518		
**Sedentary job**	0.97 (0.53–1.79)	0.925		
**Other job (non-sedentary)**	1.56 (1.16–2.10)	0.004	1.45 (1.07–1.96)	0.017
**Use of drugs**	1.38 (0.77–2.47)	0.275	1.76 (0.96–3.21)	0.066

## Data Availability

Data are available as a coded dataset.

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
