# Peer review of "Use of Drugs and Dietary Supplements in University Students of Sports Science: Results of a Survey-Based Cross-Sectional Study"

_nutrients, 2022, doi:10.3390/nu14204267_

Round 1

Reviewer 1 Report (Previous Reviewer 1)

The modifications made improved the manuscript. The authors still need to add the manuscript conclusion at the end of the discussion. 

They provided an answer in the response letter that a conclusion has been implemented, but no action was taken or marked in the manuscript

Author Response

The modifications made improved the manuscript. The authors still need to add the manuscript conclusion at the end of the discussion. 

We thank the Reviewer and we added the conclusion that have been erroneously removed from the latest updated version 

The main purpose of this study was to evaluate whether physical activity influences the use of drugs and dietary supplements. I think it cannot be accepted based on the author's current revisions. The most critical reasons are as follows:

  1. In accordance with the article's primary aim, the authors compared the use of drugs and dietary supplements among students based on subjects' different physical activity levels (PAL). But in the latest version, what are active students and agonist athletes are still not defined. Quantitative standards are needed to measure PAL (eg MET-h/week). The article's current evaluation of these two groups of people is: "agonist athletes: doing moderate or high-intensity training" and "active students: on the basis of their answers". In terms of research results citation, there is no reference value.

We are sorry that this was not clear, but the agonist athlete are those performing competitive sport and we did not find in the paper this statement "agonist athletes: doing moderate or high-intensity training", in fact in the first lines of the Results was already indicated the population of physically active (1075) and of these 371 did a competitive sport (identifying the group of agonist athletes). Regarding MET a post-hoc assessment was performed on the basis of type of activity, minutes of training and number of week days, all agonist athletes have been assigned to a MET score. The result of this assessment has been inserted in the results.

  1. In the first edition of the reviewer comments, we asked why the study population ( undergoing bachelor's degree) had a high proportion (35.6%) of the working population (5.9% had sedentary work) because of the impact of work on occupational PA. The author's reply to us is: based on the personal assessment of the participant. This doesn't help our problem. We believe this study is meaningful as a nutritional guidance program at the author's school but has no reference value for other populations outside this school. We are sorry for the misunderstanding but the phase “based on the self assessment …” was referred only to the definition of sedentary work,  we do have a high number of students that work because usually most of our students are already into sport as agonist athletes (which in Italian defines those who do competitions) or as coaches in gyms or as personal trainers. Considering the impact of work on physical activity we indeed ask if the work was sedentary or not, but our internal ethics committee did not allow us to ask specifically what kind of job was done because of privacy reasons and to maintain anonymity. We clarified these points in the paper.

In general, we believe that the study population of this study is not representative, and the authors have not provided us with sufficient and evidence-based data to provide more value for the database.

We hope that now the paper is more suitable for publication

Reviewer 2 Report (Previous Reviewer 2)

The main purpose of this study was to evaluate whether physical activity influences the use of drugs and dietary supplements. I think it cannot be accepted based on the author's current revisions. The most critical reasons are as follows:

  1. In accordance with the article's primary aim, the authors compared the use of drugs and dietary supplements among students based on subjects' different physical activity levels (PAL). But in the latest version, what are active students and agonist athletes are still not defined. Quantitative standards are needed to measure PAL (eg MET-h/week). The article's current evaluation of these two groups of people is: "agonist athletes: doing moderate or high-intensity training" and "active students: on the basis of their answers". In terms of research results citation, there is no reference value.
  2. In the first edition of the reviewer comments, we asked why the study population ( undergoing bachelor's degree) had a high proportion (35.6%) of the working population (5.9% had sedentary work) because of the impact of work on occupational PA. The author's reply to us is: based on the personal assessment of the participant. This doesn't help our problem. We believe this study is meaningful as a nutritional guidance program at the author's school but has no reference value for other populations outside this school.

In general, we believe that the study population of this study is not representative, and the authors have not provided us with sufficient and evidence-based data to provide more value for the database.

Author Response

We thank Reviewer 2 for his/her comments and we have inserted as requested the Conclusion.

Round 2

Reviewer 2 Report (Previous Reviewer 2)

can be received

This manuscript is a resubmission of an earlier submission. The following is a list of the peer review reports and author responses from that submission.

Round 1

Reviewer 1 Report

 Title:

·         I believe that the authors want to refer to dietary supplements, rather than natural supplements.

·         I also advise putting the specific study population (students attending the Bachelor Degree in Sport and Health Science) in the title instead of physically active young adults. As they are evaluating and showing the results of active and non-active participants.

Abstract:

·         I suggest writing a structured abstract

·         The objective of the study is missing

·         Specify what are natural supplements

·         Results regarding logistic regression are missing

·         Agonist athletes is not a usual term. What do the authors mean by that?

·         Which were the most used drugs?

·         Lifestyle was not assessed in this study. So it is not possible to conclude that more physically active students have a healthier lifestyle.

Introduction

·         Drug use was not covered in the introduction, only the use of supplements.

Methods

·         Did not give details of sampling. How the sample was defined, and what were the inclusion and exclusion criteria? Is it a representative sample, is it convenient? The authors need to provide more details.

·         Section “2.1 Participants” contains information on the outcome of the study, not on the methods.

·         What is the ethics committee approval protocol number?

·         What parameters were used to classify participants into active and inactive?

·         Line 84 – in the question “Use of supplements during physical activity”, the correct term would be physical exercise, instead of physical activity.

·         Socioeconomic profile was not investigated? It is an important point to be taken into account, as it is a characteristic that interferes with the use of supplements and drugs. The same for lifestyle habits, alcoholism, smoking, comorbidities?

·         What parameters were used as the outcome variable of the regression models? Use of drugs/supplements in the last 7 days?

Results

·         Add “years” to age on line 106

·         What has been established as sedentary work?

·         In line 108, when it is reported that 74% of the students performed physical activity, do the authors mean programmed recreational physical exercise? It is better to use the nomenclature correctly.

·         The 276 participants who did not respond about frequency, was it just about the frequency of training or did they not respond to the IPAQ? 26% is a high non-response rate. Could this not have affected the analyses?

·         Line 111. Are you considering as usual supplement consumers those who used them during the last 7 days or had any other questions that were not specified in the methods? According to what was written in the methods, the questions were about the use of supplements during physical activity and in the last 7 days.

·         Lines 111-112 Multivitamin integrators is not a usual term

·         Lines 112 – 178 consumed supplements in the last 7 days. And the other 27 considered as usual consumers? When was this used?

·         Weight and height results were not demonstrated.

·         Table 1 - The question “Do you usually use supplements/drugs?” was asked. It is not described in the methods.

·         Were individuals who did not respond to questions about supplement and drug use considered in the regression analyses? If so, they should be removed from the sample, as they are the outcome variables. And this must be explicit in the text.

·         In Figure 1, were all physically active individuals included or only those who practised competitive sports? Review the title of the Figure, as it was confusing.

·         Are Tables 2 and 3 referring to univariate or multivariate regressions?

·         The title of tables and figures is not self-explanatory, it needs improvement.

·         Line 147 – “the only predictor that significantly reduced drug intake”, actually the predictor reduced the chance of consuming drugs and did not reduce consumption.

·         Line 161 – the term “natural supplements” appears. What is considered natural supplements? I believe you are referring to dietary or food supplements.

·         In the introduction, the authors report that they “investigated in this specific population the possible concomitant use of supplements and drugs”. However, data on the concomitant use of drugs and supplements were not presented.

Discussion

·         Line 182 – controlled population. I do not believe you can use the term controlled in this case.

·         Line 195. The author mentioned a “healthier lifestyle”, however they did not assess lifestyle, to conclude this.

·         Line 203 – 204. Provide a reference to the first sentence of this paragraph.

·         There are other important limitations: the data is self-reported and memory dependent. Lack of data on socio-economic profile, lifestyle, and health condition that may impact drug and supplement consumption.

·         What is the conclusion of the study?

References

·         References 1 and 2 are in a different font than the others.

Author Response

thank you for your time in revising the manuscript we submitted and for the comments that helped us in ameliorating several aspects of our paper.

As suggested by the Reviewer we performed the following changes:

Title:

  •  I believe that the authors want to refer to dietary supplements, rather than natural supplements. This has been changed
  • I also advise putting the specific study population (students attending the Bachelor Degree in Sport and Health Science) in the title instead of physically active young adults. As they are evaluating and showing the results of active and non-active participants. This has been changed

Abstract:

  • I suggest writing a structured abstract We did not find specific indication on this point in the instructions for authors
  • The objective of the study is missing This has been added
  • Specify what are natural supplements This has been specified
  • Results regarding logistic regression are missing This has been added
  • Agonist athletes is not a usual term. What do the authors mean by that?This has been specified
  • Which were the most used drugs? This has been specified
  • Lifestyle was not assessed in this study. So it is not possible to conclude that more physically active students have a healthier lifestyle. This has been changed

Introduction

  • Drug use was not covered in the introduction, only the use of supplements. This has been implemented

Methods

  • Did not give details of sampling. How the sample was defined, and what were the inclusion and exclusion criteria? Is it a representative sample, is it convenient? The authors need to provide more details. This has been implemented
  • Section “2.1 Participants” contains information on the outcome of the study, not on the methods. This has been implemented
  • What is the ethics committee approval protocol number? This has been added
  • What parameters were used to classify participants into active and inactive? This has been implemented
  • Line 84 – in the question “Use of supplements during physical activity”, the correct term would be physical exercise, instead of physical activity. This has been changed
  • Socioeconomic profile was not investigated? It is an important point to be taken into account, as it is a characteristic that interferes with the use of supplements and drugs. The same for lifestyle habits, alcoholism, smoking, comorbidities? This cannot be implemented because we couldn't ask these questions and has been reported as a limit in the discussion 
  • What parameters were used as the outcome variable of the regression models? Use of drugs/supplements in the last 7 days? This has been implemented

Results

  • Add “years” to age on line 106 This has been added
  • What has been established as sedentary work? This has been implemented
  • In line 108, when it is reported that 74% of the students performed physical activity, do the authors mean programmed recreational physical exercise? It is better to use the nomenclature correctly. This has been implemented
  • The 276 participants who did not respond about frequency, was it just about the frequency of training or did they not respond to the IPAQ? 26% is a high non-response rate. Could this not have affected the analyses? We did specify in the methods how many responded to all the questions
  • Line 111. Are you considering as usual supplement consumers those who used them during the last 7 days or had any other questions that were not specified in the methods? According to what was written in the methods, the questions were about the use of supplements during physical activity and in the last 7 days. This has been implemented
  • Lines 111-112 Multivitamin integrators is not a usual term This has been changed
  • Lines 112 – 178 consumed supplements in the last 7 days. And the other 27 considered as usual consumers? When was this used? This has been implemented
  • Weight and height results were not demonstrated. This was not inserted because was not reported by most of the subjects and could represent a bias
  • Table 1 - The question “Do you usually use supplements/drugs?” was asked. It is not described in the methods. This has been implemented
  • Were individuals who did not respond to questions about supplement and drug use considered in the regression analyses? If so, they should be removed from the sample, as they are the outcome variables. And this must be explicit in the text. This has been implemented
  • In Figure 1, were all physically active individuals included or only those who practised competitive sports? Review the title of the Figure, as it was confusing. This has been implemented
  • Are Tables 2 and 3 referring to univariate or multivariate regressions? This has been implemented
  • The title of tables and figures is not self-explanatory, it needs improvement. This has been implemented
  • Line 147 – “the only predictor that significantly reduced drug intake”, actually the predictor reduced the chance of consuming drugs and did not reduce consumption. This has been implemented
  • Line 161 – the term “natural supplements” appears. What is considered natural supplements? I believe you are referring to dietary or food supplements. This has been changed into dietary supplements
  • In the introduction, the authors report that they “investigated in this specific population the possible concomitant use of supplements and drugs”. However, data on the concomitant use of drugs and supplements were not presented. This has been implemented

Discussion

  • Line 182 – controlled population. I do not believe you can use the term controlled in this case. This has been changed
  • Line 195. The author mentioned a “healthier lifestyle”, however they did not assess lifestyle, to conclude this. This has been changed
  • Line 203 – 204. Provide a reference to the first sentence of this paragraph. This has been implemented
  • There are other important limitations: the data is self-reported and memory dependent. Lack of data on socio-economic profile, lifestyle, and health condition that may impact drug and supplement consumption. This has been implemented
  • What is the conclusion of the study? This has been implemented

References

  • References 1 and 2 are in a different font than the others. This has been changed

Author Response

We want to thank the Reviewer for the time dedicated to our paper and we believe that now the manuscript has been implemented thanks to his/her comments.

We performed the changes accordingly to the suggestions.

Abstract 

1) The abstract should be reworded. This has been implemented

2) This background information was too long for the abstract and didn’t fully explain the reasoning for the study. Also, no hypothesis or aim is written in the abstract. This has been implemented

3) Ln 19-20 Active, Agonist are adjectives, how to define or identify them? The description of the participants should be more quantitative and detailed, especially when used for stratified. This has been implemented

4) Ln 21 What do you mean with the “higher number”? Any statistical difference? This has been implemented

5) Ln 25-26 The conclusion should be reworded. Neither of the conclusions can be concluded through your study. Again, clarify what is physically active. Besides, too vague to define healthier lifestyle. The results of the present study should be explained more cautiously.  This has been implemented

Introduction 

1) The rationale of the present study is not clear. The introduction should be revised substantially.  This has been implemented

2) Author devoted considerable space to describing sources of information on nutritional supplementation; assessment of supplementation needs and sources of purchasing supplements. However, no hypothesis is written in the introduction. That is remarkably deficit since the authors have to help the reader to understand why the survey is valuable and essential.  This has been implemented

3) What is the new contribution of the present study to the literature? What is the significance of the present study?  This has been implemented

4) Only nutritional supplements were mentioned in the introduction. In the title, there were including Drugs and Natural Supplements. Please clarify what natural supplements are and did all the supplements mentioned in your study were natural supplements? Besides, how drug use may affect health and exercise performance should be added and clarify reasons, time periods and types of drug use.  This has been implemented

5) Ln 57-58 What do you mean with this sentence? What does this sentence have to do with the purpose of the study? Insufficient rationale for selecting study population. 

This has been implemented

Methods 

1) More details should be provided.  This has been implemented

2) When was the survey conducted? Many factors like whether or epidemic disease may affect physical activity level and the use of nutritional supplements and drugs. Please clarify the time period you collected all the data.  This has been implemented

3) Please clarify the definition of active in doing ** intensity training and agonist athletes. No Inclusion and exclusion criteria were provided, please add more details and list more specific criteria of the participants. The health conditions are strongly related to the drug and supplements use.  This has been implemented

4) In the abstract, you mentioned a modified version of IPAQSF was used. Clarify what modifications have been made to the questionnaire and whether the modified version was validated?  This has been implemented

5) How many questionnaires have you sent out and how many have been returned? In addition to the reliability and validity of the questionnaire itself, how do you judge whether the questionnaire you returned is valid? You need to provide these information in the manuscript.  This has been implemented

6) Ln 78 You referred the survey was self-administered, how to ensure the quality of questionnaires?  This has been implemented

7) Provide more details of the questionnaire. Only drug use condition was mentioned? Nothing about health conditions? It’s normal to take pills when getting a cold and there is nothing to do with PA level or knowledge awareness. 

This has been implemented

8) Besides the questionnaire do you collect other information or any other measurements that have been conducted? 

This has been implemented

Results 

1) Ln 107 You mentioned some of your participants were workers and some have secondary jobs but in the methods you referred your participants were students. Confusing. 

This has been implemented

2) Ln 111How to define usual supplement users, please clarify.  This has been implemented

3) Ln 113 How to define regular drug user, please clarify.  This has been implemented

4) Figures need improvement. This has been implemented

5) Table 2 Do you take health conditions into consideration? Or do you have these information which was quite important.  This has been implemented

6) Since some of them have jobs, besides exercise training, was labor-based physical activity considered?  This has been implemented

Disscussion 

1) It is as a reader difficult to follow the built up to the purpose of study and the discussion of the results compared to others since the language is not precise and important details are missing.  This has been implemented

2) Ln 198-202 You have mentioned similar study in the paragraph discussion. hat is the new contribution of the present study to the literature? These should be highlighted and discussed in the discussion section This has been implemented. 

Round 2

Reviewer 2 Report

The authors have made great efforts to revise the manuscript. According to the updated version, there are still few explanations of the rationale for study design in general. As the author admitted, there was lack of health conditions investigations which were quite important and the study was conducted in June and July 2021, when Covid-19 may affect PA levels which should also be discussed. Besides, whether the selection of the subjects is appropriate remains to be discussed according to the author's description.

Author Response

According to the Reviewer's comments we performed changes to the introduction to better specify the aim of our study and in the discussion to clarify the other issues. We hope that now everything is in order for publication.